# Functional Comparison of Blood-Derived Human Neural Progenitor Cells

**DOI:** 10.3390/ijms21239118

**Published:** 2020-11-30

**Authors:** Eszter Szabó, Flóra Juhász, Edit Hathy, Dóra Reé, László Homolya, Zsuzsa Erdei, János M. Réthelyi, Ágota Apáti

**Affiliations:** 1Institute of Enzymology, Research Centre for Natural Sciences, 1117 Budapest, Hungary; szabo88eszter@gmail.com (E.S.); juhaszflora125@gmail.com (F.J.); daura.ree@gmail.com (D.R.); homolya.laszlo@ttk.mta.hu (L.H.); erdeizs@gmail.com (Z.E.); 2Department of Psychiatry and Psychotherapy, Faculty of Medicine, Semmelweis University, 1083 Budapest, Hungary; hathyedit@gmail.com; 3National Brain Research Project (NAP) Molecular Psychiatry Research Group, Hungarian Academy of Sciences and Faculty of Medicine, Semmelweis University, 1083 Budapest, Hungary

**Keywords:** neural progenitors, interrupted reprogramming, iPSC, neural differentiation, schizophrenia

## Abstract

Induced pluripotent stem cell (iPSC)-derived neural progenitor cells (NPCs) are promising tools to model complex neurological or psychiatric diseases, including schizophrenia. Multiple studies have compared patient-derived and healthy control NPCs derived from iPSCs in order to investigate cellular phenotypes of this disease, although the establishment, stabilization, and directed differentiation of iPSC lines are rather expensive and time-demanding. However, interrupted reprogramming by omitting the stabilization of iPSCs may allow for the generation of a plastic stage of the cells and thus provide a shortcut to derive NPSCs directly from tissue samples. Here, we demonstrate a method to generate shortcut NPCs (sNPCs) from blood mononuclear cells and present a detailed comparison of these sNPCs with NPCs obtained from the same blood samples through stable iPSC clones and a subsequent neural differentiation (classical NPCs—cNPCs). Peripheral blood cells were obtained from a schizophrenia patient and his two healthy parents (a case–parent trio), while a further umbilical cord blood sample was obtained from the cord of a healthy new-born. The expression of stage-specific markers in sNPCs and cNPCs were compared both at the protein and RNA levels. We also performed functional tests to investigate Wnt and glutamate signaling and the oxidative stress, as these pathways have been suggested to play important roles in the pathophysiology of schizophrenia. We found similar responses in the two types of NPCs, suggesting that the shortcut procedure provides sNPCs, allowing an efficient screening of disease-related phenotypes.

## 1. Introduction

Human neuronal cell cultures are essential for biological and preclinical studies of our nervous system. However, the limited access of primary human adult neural samples and the severe limitations of primary neuronal cultures (such as donor to donor variability and restricted proliferative capacity) make the interpretation and standardization of data extremely difficult. Recent developments in cell reprogramming represent a unique opportunity to generate in vitro models for the human nervous system. In essence, there are three different approaches to derive neural cultures from adult somatic cells: full reprogramming of somatic cell types followed by directed differentiation, partial or interrupted reprogramming of fibroblasts, or blood progenitors combined by specialized culture conditions (also called cell-activation and signaling-directed lineage conversion (CASD)) and direct conversion of fibroblast into neurons by the overexpression of special transcription factors. Using a full reprogramming technique, neural cultures can be generated from all nucleated human somatic cell types through the formation of stable induced pluripotent stem cells (iPSCs). Using a wide variety of gene delivery methods, forced expression of certain transcription factors (usually the classical Yamanaka factors—octamer-binding transcription factor 3/4 (Oct3/4), SRY (sex determining region Y)-box 2 (SOX2), Kruppel-like factor 4 (Klf4), and avian myelocytomatosis virus oncogene cellular homolog (c-Myc)) results in pluripotent cell cultures (reviewed in [1]). Next, multiple neuronal subtypes can be differentiated from iPSCs by various protocols, i.e., specific morphogenic patterning cues. Apart from the undeniable advantages of this technique, it also has disadvantages—it is labour intensive, cost- and time-consuming, and requires special expertise. On the other hand, a direct conversion of fibroblasts or human adult peripheral blood T cells [2] to neuronal cells can be achieved using the overexpression of neuronal transcription factors. In this approach, first Achaete-scute homolog 1 (Ascl1), Brain2 (Brn2), and myelin transcription factor (Myt1l) were used [3], and then a wide variety of factors and molecules were applied (reviewed in [4]) to produce induced neurons (iN) in a simpler and shorter process (taking only weeks). Moreover, the iNs have the potential to maintain the aging and epigenetic signatures of the donor. Nevertheless, to assess the maturation status of iNs, the selection of the generated neurons and the scalability of the method raise unique challenges. The interrupted reprogramming overcomes some of these aforementioned problems (see [5]). Neural progenitor cell (NPC) cultures were established from fibroblasts or blood progenitor cells by starting the reprogramming protocol with delivery of classical Yamanaka factors, using nonintegrating methods. Thereafter, before reaching a pluripotent state, the cells were further differentiated by changing the culture conditions into neurogenic conditions (CASD), with or without growth factor supplementation [6]. These methods provided homogeneous and proliferative NPC populations, and the devoted cost and time decreased dramatically by bypassing the iPSC derivation process. The generated NPCs could be further differentiated toward various neural cell types, e.g., neurons, astroglia, and oligodendrocytes, in specified differentiation media [7,8,9]. There are growing numbers of articles reporting generation of human NPCs by different methods; however, there are no data about a detailed comparison of NPCs that were generated from the same source by using different methods. To allow such a comparison, we used Sendai virus reprogramming methods for the generation of NPCs from fully reprogrammed (classic) and partially reprogrammed (shortcut) cells, because these methods allow the generation of desired cell types effectively without transgene integration [10,11]. For the generation of NPCs, we used the same blood samples and the same Sendai virus transduction for reprogramming and performed comparative studies in terms of the RNA and protein expression profiles and various functional assays.

Four blood samples were taken from donors of different age, sex, and disease status. Three samples came from a family trio—a patient diagnosed with schizophrenia and his unaffected parents, since we also aimed to compare the disease modeling applicability of these two NPC generation methods. An unrelated umbilical cord blood (UCB) sample served as an additional control.

Schizophrenia (SCZ) is a complex neurodevelopmental disorder [12,13] affecting ~1% of the world’s population. It is highly heritable, characterized by positive, negative, and cognitive symptoms, which lead to a severe social and economic burden on the patient and the society. The experimental tools used for investigation of SCZ pathogenesis (in vivo imaging and ex vivo/postmortem human studies or animal models) resulted in valuable data but have their limitations, which can be overcome by new approaches. Reprogramming of patient-specific cells is a promising new possibility for modeling human neurodevelopment and has been used for studying the mechanisms underlying schizophrenia [14]. The iPSC-derived neural progenitors and neural cell types are immature relatives to those in the human brain, and these cells resemble fetal brain tissue; thus, they are more suited for the study of disease predisposition. Indeed, besides the investigations of patient-derived neurons (reviewed in [15,16]), several studies have investigated the patient versus control differences at early phases of neural differentiation at the NPC stage. Altered proliferation rate [17] and Wnt-signaling [18,19], differences in migration capacity and cell polarity [20], and vulnerability to oxidative stress [21] were observed. In this study, we investigated the NPCs generated from a SCZ patient and control NPCs by two different methods and compared the differences regarding cell proliferation, migration, oxidative stress, and calcium signaling.

## 2. Results

### 2.1. Donor Selection and NPC Generation

In this study, we selected four donors: a male patient (1PJ) diagnosed with SCZ and his unaffected parents (1MA—mother and 1FJ—father) and a healthy new-born female (UBC2—umbilical cord blood2). Blood samples were obtained after written informed consent. The iPSC generation process and the study were approved by the Human Reproduction Committee of the Hungarian Health Science Council (ETT HRB). Approval number: 33873-3/2014-HER. NPCs were generated from peripheral blood mononuclear cells (PBMCs) and UCB and represent three healthy and one diseased donor with a wide variety of ages (59 years to new-born) and balanced sex distribution (two males and two females). Mononuclear cells were separated from peripheral blood or umbilical cord blood and the samples were transduced by Sendai virus vectors containing transcriptional factors Sox2, Oct3/4, Klf4, and c-Myc. The transduced cells were replated on mitomycin-treated mouse embryonic fibroblasts (MEF) on the third day. When the clumps with pluripotent cell-like morphology appeared, some of them were mechanically passaged to new MEF or alternatively to Matrigel. The clumps on MEF were further cultured and cloned until stable iPSC lines were generated, whereas the cells on Matrigel were passaged to a poly-ornithine/laminin-coated surface by accutase and cultured in NPC media until homogeneous NPC lines (shortcut NPCs—sNPCs) were established after a few passages. Finally, the donor-derived iPSCs were differentiated into hippocampus-patterned NPCs (classical NPCs—cNPCs) as described previously [22,23] (Appendix A). We successfully generated proliferative NPCs from all the four donors using two different techniques, and next we started to characterize the sNPCs (made by interrupted reprogramming) and the cNPCs (derived from iPSCs). A detailed comparison of morphological and functional phenotypes between traditionally derived cNPC lines using multiple clones from members of the case-parent trio is demonstrated in [24]. This study focuses on the description of the alternative differentiation method, i.e., partial reprogramming and characterization of shortcut NPCs.

### 2.2. Characterization of NPCs Generated by Different Methods

Both sNPCs and cNPCs showed similar homogenous morphology after six passages and were studied between 6 and 18 passages. The stable, still proliferating iPSC-derived cNPC and sNPC cultures were characterized by immunostaining and mRNA expression profiling of stage-specific markers. Moreover, they were differentiated into hippocampal dentate gyrus (DG) granule neurons [22,23], and neural markers were also investigated by immunostaining.

As demonstrated in Figure 1a, the mRNA expression of the pluripotency marker Nanog was significantly lower in both sNPCs and cNPCs compared to iPSCs, while we found no significant differences between the two types of NPC-s. SOX2 is known to be expressed in both iPSCs and NPCs, its expression levels were slightly elevated in sNPCs and cNPCs but as they varied within one order of magnitude in all three cell types differences were not significant (for exceptions see Appendix A). The neural progenitor marker Nestin showed a significantly increased level in all NPCs compared to iPSCs, and in the case of 1MA and 1PJ derived cells, we also found differences between sNPSs and cNPCs but as this variation was not consistent, we assume that it is not by reason of the differentiation protocols. mRNA levels of PAX6 and FOXG1, markers for characterization of the neural differentiation, were significantly upregulated in the sNPCs and most of cNPCs except for 1FJ in the case of PAX6 and 1MA in the case of FOXG1 where the observed differences did not reach the significance level. For the p values of the statistical analysis see Appendix A.

Figure 1b and Appendix A demonstrate the morphology and immunocytochemical features of NPCs, while Figure 1c illustrates the differentiated neurons thereof. We found that both the iPSC-derived cNPCs and the shortcut sNPCs showed double positive staining for transcription factor SRY (sex determining region Y)-box 2 (SOX2) and intermediate filament Nestin in all samples. Moreover, all NPC-derived DG neural cultures showed intensive staining for microtubule-associated protein 2 (MAP2), highlighting the neuronal identity and morphology, and were partially positive for the DG-specific transcription factor prospero homeobox protein 1 (PROX1).

These findings indicate that all the generated NPCs are neural progenitors, which can be differentiated into Prox1-positive neurons. Morphological, immunocytochemical, and mRNA expression features of the cNPCs and sNPCs were similar; however, we found some variations in mRNA expression levels, especially in cNPCs, while sNPCs showed more uniform mRNA expressions.

### 2.3. Functional Comparison of Classic and Shortcut NPCs

Next, to compare the two types of NPCs and investigate the differences among NPCs derived from the SCZ patient (1PJ), from his unaffected parents (1FJ and 1MA, as controls) and from an unrelated healthy donor (UCB2), we analyzed proliferative capacity and migration ability, the oxidative stress and mitochondrial status, as well as calcium signals of both cNPCs and sNPCs.

Wnt-signaling may contribute to SCZ [25] and has been reported to be important for proliferation and migration of neural progenitors [17,19,21]. We utilized the two types of NPCs to investigate their proliferation and migration activity in the SCZ patient and in the controls.

Figure 2a shows the growth curves of different NPCs measured by flow cytometry (FCM) from day 2 to day 4 and normalized to the initial cell number (3 × 10^4^/well). The data show no significant difference between cNPCs and sNPCs (for *p* values see Appendix A). When we compared growth rates of the patient-derived NPCs to the ones that were derived from the parents or from the unrelated healthy donor at day 4 (Figure 2b and Appendix A), we concluded that the patient-derived NPCs did not show reduced cell proliferation activity.

Cell migration plays a major role in the formation and function of the central nervous system, and its alterations have been observed in disease development. Cell migration is frequently evaluated in vitro by the monolayer wound healing assay or “scratch assay” [26]. In these experiments, we wounded the confluent cell layers of NPCs mechanically and analyzed the coverage of open area by migrating and proliferating cells after 24 h using bright-field images taken by a high content screening system (Appendix A). In Figure 3a, image analysis from two independent experiments (3–3 parallels in each) shows that no significant differences were found between shortcut and classic, or between SCZ patient-specific and healthy NPCs (*p* = 0.4363 and *p* = 0.3730, respectively).

Increased oxidative stress and mitochondrial damage were observed in the case of forebrain NPCs derived from SCZ patients [21]; thus, we characterized the level of oxidative stress and the status of mitochondria in our NPCs.

We applied CellROX™ Green Reagent (Thermo Fisher Scientific, Waltham, MA, USA) for oxidative stress detection in both NPC groups in order to compare the cytosolic reactive oxygen species (ROS) level in SCZ patient-derived and control NPCs. We analyzed FCM data (Appendix A) from two independent experiments with two technical parallels each. ROS levels (Figure 3b) in the SCZ patient-derived NPCs were similar to the controls in both NPC groups, and no significant differences were found between the differently generated NPCs (*p* = 0.0756 and *p* = 0.3075, respectively).

Next, Mitotracker Red CMXRos (Thermo Fisher Scientific, Waltham, MA, USA) (a mitochondrion-specific dye) was used for the quantification of mitochondria in the NPCs. Cells were stained using the recommended protocol, and the fluorescence intensity of the dye was evaluated and normalized to cell numbers by ImageXpress High Content Screening system (Molecular Devices, LLC., San Jose, CA, USA). There were no significant differences between the NPCs (*p* = 0.0617 between donors and *p* = 0.5119 between differentiation protocols), showing the similarity of NPCs derived by the two different methods. In order to evaluate mitochondrial morphology, we analyzed confocal images (Appendix A) taken by Zeiss LSM 900, using the NetworkAnalysis (MiNA ) toolset of ImageJ (U. S. National Institutes of Health, Bethesda, MD, USA), software to determine the mitochondrial footprint, mean branch length, summed branch length, and network branches. We found no significant differences between any of these parameters as shown in Appendix A (for *p* values see Appendix A).

Hypofunction of glutamate-triggered signaling has been implicated in SCZ [27], and pharmacological modulation of the glutamate pathway is one of the future possibilities to treat patients with SCZ [28].

To examine the glutamate signaling in our model, we performed a detailed characterization of the glutamate-induced calcium signals using Fluo-4 (Thermo Fisher Scientific, Waltham, MA, USA) loaded NPC cultures. Figure 4a shows typical calcium signals obtained after treating the cells with 100 µM glutamate, followed by 50 mM KCl. As documented, after the addition of glutamate, all types of NPCs responded with well-measurable calcium signals, while the cells did not show significant calcium response after KCl treatment. It has been noted previously [23] that NPCs do not respond to KCl, and only the partially differentiated cells (>10% of the NPC cultures) show a weak signal in this case. Therefore, these cells were excluded from further analysis. We analyzed hundreds of cells (Figure 4b) and found no significant differences in the number of responding cells or signal intensities after addition of glutamate when comparing the patient-specific NPCs to NPCs derived from his parents (Figure 4c–e). The only statistically significant alteration was seen in the case of UCB2-derived classic NPCs, where a steeper increase of the Ca-signal was observed (Figure 4a,c and Appendix A).

## 3. Discussion

New biologically relevant screening models can facilitate the discovery of better treatments for neuropsychiatric disorders. However, the physiological relevance of hPSC-derived neural cell types remains a subject of intense debate [29]. Therefore, a better understanding of iPSC-derived neural models and making them more accessible are critical in making this model system suitable for disease modelling and drug screening. Generation of homogeneous and functional neurons from iPSCs is cost- and time-consuming; thus, the methods providing faster and less expensive solutions improve the effectiveness of drug screening studies. Application of neural progenitor cells for disease modeling and drug screening is an emerging field, and its applicability has been proven for neurodevelopmental diseases [14,30,31]. In this work, we generated sNPCs (by interrupted reprogramming/CASD), and compared them to iPSC-derived NPCs, in order to find their similarities or potential differences.

Neural cells are generated typically from fibroblasts; however, the long-term in vitro culturing of fibroblasts before reprogramming may facilitate the occurrence of mutations. Using peripheral blood cells for full reprogramming is a generally applied method, while currently there are only limited data describing the generation of neural cells from blood by bypassing the pluripotent state. Direct neuronal conversion protocols use mainly fibroblasts [4], and interrupted reprogramming (or CASD) methods use neural cells from fibroblasts or hematopoietic stem cells [14] as starting material. In this study, PBMCs were separated from blood samples and cultured only for 2 days to avoid genetic alterations before reprogramming. Moreover, we did not use any selection for hematopoietic stem cell markers, further decreasing cell culturing time compared to other methods. With this method, we minimized the possibility of genetic alterations before reprogramming.

We have successfully generated NPCs with proliferative and differentiation capacity from all four donors’ PBMCs, using the same blood samples and two different techniques. Both types of NPCs can be propagated to at least passage 30 without losing their proliferative capacity and providing sufficient cell quantities for functional assays and drug screening applications. We found no major variances in the efficiency of NPC generation techniques, independently of age, sex, disease status, or the method of generation. Moreover, no consistent differences in the differentiation efficacy among different NPC lines were detected.

When we compared the proliferation of different NPCs (iPSC-derived classic and shortcut cells or patient-derived to healthy controls) we found no significant differences. The scratch assay showed similar results in the pairwise comparison between cNPCs and sNPCs and that NPCs derived from the patient’s sample did not differ from those obtained from the control samples. The scratch assay showed a combination of proliferation and migration of NPCs, although the quantification of the open area was manually performed and was rather subjective. Therefore, we also performed a neurosphere outgrowth assay [21] and found no significant differences between the various NPCs (data not shown). In the other functional assays, i.e., Ca^2+^-signaling and analysis of oxidative status of NPCs, we experienced no significant differences between NPCs generated by the two different methods, and we found no significant alterations between SCZ patient-derived and control NPCs, shown in earlier studies [19,21,27]. SCZ is an extremely complex disease with a variable genetic and epigenetic background, and finding an in vitro phenotype is particularly difficult.

Our results showed that classic and shortcuts NPCs have similar characteristics, and the shortcut NPCs can be properly used for monitoring in vitro phenotypes. In this study, we demonstrated that bypassing the iPSC state reduces the cost and time and thus makes the analysis feasible for a higher number of samples.

Future perspectives: Only a small number of large-scale drug screens using stem cell-based neural cultures have been published thus far, and all of these studies used hPSC-derived NPCs or neurons (reviewed in [32]). As we demonstrated here, a shorter and lower-cost method results in NPCs similar to the iPSC-derived ones and opens the possibility of more effective, personalized drug screening and disease modeling.

## 4. Materials and Methods

### 4.1. Cell Culturing and Differentiation

The iPSCs were established from a male schizophrenia patient, who is a carrier of 3 nonsynonymous de novo mutations in genes leucine-rich repeat containing 7 (LRRC7 (1:70505093G>A)); KH-type splicing regulatory protein (KHSRP (19:6416869C>A)); and killer cell immunoglobulin-like receptor, two domains, long cytoplasmic tail 1 (KIR2DL1(19:55286658A>T)), as well as from his unaffected parents [24]. The unrelated healthy control (UCB2) cells were not studied for these specific genetic mutations. iPSC lines were generated from freshly isolated PBMCs (using BD Vacutainer CPT Cell Preparation Tubes with Sodium HeparinN. REF 362753) of healthy and diseased individuals by Sendai virus (Thermo Fisher Scientific, Waltham, MA, USA)-based reprogramming and were maintained on mitomycin-C (Sigma, St. Louis, MO, USA)-treated MEF cells (Millipore, Billerica, MA, USA) until passage p15. iPSC cells were then transferred to Matrigel (Corning, New York, NY, USA)-coated plates in mTeSR™ medium (Stemcell Technologies, VAN, Canada) and were cultured to high density.

“Classical” neural progenitor cells (cNPCs) were differentiated from stabile iPSC lines as described previously [22] by Yu et al. (2014). On day 1, the cells were detached with collagenase (Thermo Fisher Scientific, Waltham, MA, USA) and transferred to ultra-low attachment plates (Nalgene Nunc International, New York, NY, USA) to allow embryoid body (EB) formation. On day 2, the medium was changed to DMEM/F-12, GlutaMAX™ (Thermo Fisher Scientific, Waltham, MA, USA) medium supplemented with N2/B27 (Thermo Fisher Scientific, Waltham, MA, USA), anticaudalizing agents Noggin (500 ng/mL; Thermo Fisher Scientific, Waltham, MA, USA), Dickkopf-related protein 1 DKK1 (500 ng/mL; PeproTech, Rocky Hill, NJ, USA), cyclopamine (1 μg/mL; Merck, Darmstadt, Germany), and SB431542 (4 μg/mL; Sigma, St. Louis, MO, USA). The cells were kept in EB culture for 20 days while medium was changed every other day. On day 21, the EBs were moved to adherent conditions on poly-ornithine (Sigma, St. Louis, MO, USA)-/laminin (Thermo Fisher Scientific, Waltham, MA, USA)-coated plates and the medium was changed to DMEM/F-12, GlutaMAX N2/B27 medium containing laminin (1 μg/mL). These conditions supported further differentiation, indicated by rosette formation. From day 27, manually picked rosettes were dissociated by Accutase (Thermo Fisher Scientific, Waltham, MA, USA) and reseeded onto new polyornithine-/laminin-coated plates in DMEM/F-12, GlutaMAX N2/B27 medium containing fibroblast growth factor 2 (FGF2; 20 ng/mL; Thermo Fisher Scientific, Waltham, MA, USA) and laminin (1 μg/mL) and Rock inhibitor (Y27632, 0.1 µL/mL; Selleck Chemicals, Houston, TX, USA). The attached NPCs were further cultured under these conditions but without Rock inhibitor and showed morphological homogeneity after 5 passages.

Shortcut neural progenitor cells (sNPCs) were differentiated from Sendai virus-transduced PBMCs before stabilizing iPSC state. Colonies from mouse embryonic fibroblast feeder cells were picked and transferred onto to Matrigel-coated plates in mTeSR medium on days 7–10. They were passaged twice on Matrigel with Accutase, and then were transferred onto polyornithine-/laminin-coated plates and were cultured as cNPCs. Both sNPCs and cNPCs could be propagated up to 30 passages, when they started to show signs of senescence (i.e., slower proliferation rate and altered morphology). cNPCs between passages p6 and p18 and sNPCs between passages p10 and p18 were used for the experiments.

NPCs were further differentiated to neurons, as described previously [22]. Prior to this culturing, we increased the temperature to 38.5 °C for 5 days in order to get rid of residual Sendai viral vectors. 1–1.2 × 10^4^ cells were seeded onto polyornithine-/laminin-coated 8-well Nunc Lab-Tek II Chambered Coverglass (Nalgene Nunc International, New York, NY, USA). On day 1, medium was changed in DMEM/F-12, GlutaMAX N2/B27 supplemented with ascorbic acid (200 ng/mL; Sigma, St. Louis, MO, USA), brain-derived neurotrophic factor (BDNF; 20 ng/mL; PeproTech, Rocky Hill, NJ, USA), cyclic adenosine monophosphate (cAMP) (500 μg/mL; Sigma, St. Louis, MO, USA), laminin (1 μg/mL), and Wnt3A (20 ng/mL; Research and Diagnostic Systems Inc., Minneapolis, MN, USA). After 1–2 weeks, when cells started to fill up the surface, we omitted Wnt3A from the media. The medium was changed every other day for 4–5 weeks.

### 4.2. Immunocytochemical Staining

For immunofluorescence staining of NPCs, we plated 3 × 10^4^ cells on polyornithine-/laminin-coated 8-well chambers. The next day, cells were fixed with 4% paraformaldehyde (PFA; Thermo Fisher Scientific, Waltham, MA, USA) in Dulbecco’s modified phosphate-buffered saline (DPBS; Sigma, St. Louis, MO, USA) for 15 min at room temperature. After the samples were washed with DPBS, we blocked them for 1 h at room temperature in DPBS containing 2 mg/mL bovine serum albumin (BSA; Sigma, St. Louis, MO, USA), 1% gelatine from cold water fish skin (Sigma, St. Louis, MO, USA), 5% goat serum (Sigma, St. Louis, MO, USA), and 0.1% Triton × 100 (Sigma, St. Louis, MO, USA). The samples were then incubated for 1 h at room temperature with the following antibodies: anti-SOX2 (SRY (sex determining region Y)-box 2)) (monoclonal/mouse, 1:20 dilution; MAB2018, R&D Systems, Minneapolis, MN, USA) and anti-Nestin (polyclonal/rabbit, 1:250 dilution; ab92391, Abcam, Cambridge, UK). After the cells were washed with DPBS, we incubated them for 1 h at room temperature with Alexa Fluor 488-conjugated goat anti-mouse immunoglobulin G (IgG) or Alexa Fluor 643-conjugated goat anti-rabbit IgG secondary antibodies (Thermo Fisher Scientific, Waltham, MA, USA). The nuclei were stained with DAPI (2-(4-amidinophenyl)-1H-indole-6-carboxamidine, Thermo Fisher Scientific, Waltham, MA, USA).

For immunofluorescence staining of neurons, we fixed 4–5-week-old differentiated cultures, as described previously. Samples were blocked for 1 h at room temperature in DPBS containing 5% goat serum and 0.1% Triton × 100. The samples were then incubated overnight at 4 °C with the following antibodies: anti-PROX1 (polyclonal/rabbit, 1:500 dilution; ab101851, Abcam, Cambridge, UK) and anti-MAP2 (monoclonal/mouse, 1:500 dilution; M1406, Sigma/Merck, Darmstadt, Germany). After cells were washed with DPBS, we incubated them for 1 h at room temperature with appropriate secondary antibodies: Alexa Fluor 633-conjugated goat anti-mouse IgG or Alexa Fluor 488-conjugated goat anti-rabbit IgG. The nuclei were counterstained with DAPI.

The stained samples were examined by a Zeiss LSM 900 confocal laser scanning microscope. Blue and green fluorescence images were acquired sequentially between 440–470 and 505–525 nm at 405 and 488 nm excitations, respectively, whereas far red fluorescence was acquired above 645 nm at 633 nm excitation. Fluorescence images were analyzed with ImageJ software (Fiji https://doi.org/10.1038/nmeth.2019). In the confocal images, pseudo-color coding was used for better visualization.

### 4.3. Gene Expression Analysis

Total RNA was isolated from iPSCs and NPCs using TriFast reagent, following the manufacturer’s instructions (Peqlab Ltd., Fareham, United Kingdom). Complementary DNA (cDNA) samples were prepared from 1 μg total RNA using the Promega Reverse Transcription System Kit (Promega, Madison, WI, USA) as specified by the manufacturer. For real-time quantitative PCR (RT-PCR), we purchased the following Pre- Developed TaqMan assays from Thermo Fisher Scientific (Waltham, MA, USA): NANOG as undifferentiated stem cell marker, SOX2 as undifferentiated and NPC marker, NESTIN as marker of NPC state, PAX6 and FOXG1 as neural differentiation specific markers, and RPLP0 ribosomal protein as endogenous control for quantification. RT-PCR analyses were carried out using the StepOnePlus Real-Time PCR System (Thermo Fisher Scientific, Waltham, MA, USA), according to the manufacturer’s instructions. The changes in mRNA levels were determined by the 2^−ΔCt^ method using RPLP0 as endogenous control gene. Relative mRNA levels of 2 independent experiments (3 technical replicates each) were analyzed by one-way ANOVA statistics.

### 4.4. Growth Curve Measurements

To measure growth curves, we transferred cells to 24-well plates with densities of 3 × 10^4^ cells per well. From day 2 to day 4, we detached each day 3 parallel wells by Accutase, and resuspended them in 0.5% BSA in phosphate-buffered saline (PBS). Cell numbers were calculated in Attune NxT flow cytometer using propidium iodide (Thermo Fisher Scientific, Waltham, MA, USA) to eliminate dead cells from analysis. N/N0 values were calculated and compared for each day, where N is the cell count on a particular day and N0 is the number of cells seeded on the well originally.

### 4.5. Scratch Test

A total of 400,000 NPCs were seeded onto poly-ornithine/laminin coated, 6-well plates. After reaching confluence, scars were made by a 5 mL pipette in each well in triplicates. To analyze scratch closure, bright field images were taken using ImageXpress High Content Screening system (Molecular Devices, LLC., San Jose, CA, USA) at 0 and 24 h. We compared average percentage of scratch closure on two biological parallel experiments (3–3 technical parallels in each).

### 4.6. ROS Test

Baseline levels of reactive oxygen species (ROS) were investigated using the CellROXGreen kit (Thermo Fisher Scientific, Waltham, MA, USA) following the manufacturer’s instructions. NPCs were dissociated by accutase, and 100,000 cells were incubated with CellROX Green reagent at 500 nM final concentration for 30 min at 37 °C. Then, cells were washed with 1 × PBS. The mean fluorescence signal of labelled cells was measured by FCM (Attune NxT Flow Cytometer, Thermo Fisher Scientific, Waltham, MA, USA), before measurement propidium iodide (Thermo Fisher Scientific, Waltham, MA, USA) was applied to gate out the dead cells. The medians of green fluorescence intensity were compared in the populations unstained for propidium iodide in two biological parallel experiments (3–3 technical parallels in each).

### 4.7. Mitotracker Assay

Mitochondrial function of NPCs was measured by Mitotracker Red dye (MitoTracker™ Red CMXRos, Cat. Number: M7512, Thermo Fisher Scientific, Waltham, MA, USA). NPCs were plated onto 8-well chambers previously coated with poly-ornithine/laminin. After reaching confluence, live cells were incubated with 250 nM Mitotracker Red dye for 30 min at 37 °C. The dye was then washed with DPBS, and the NPCs were fixed by 4% PFA for 15 min at RT. After fixation, nuclei were stained with 1 µM DAPI. Fluorescent signals were acquired by ImageXpress High Content Screening system (Molecular Devices, LLC., San Jose, CA, USA) and confocal microscopy (Zeiss LSM 900). By the high content screening system, Fluorescent images were taken on the whole surface of the chambers, then at least 25 frames from each well were selected for further evaluation. Integrated dye intensities relative to cell count in the particular frame were calculated and compared from two parallel stainings. For mitochondrial morphology analysis, confocal images were evaluated for each cell lines using the Mitochondrial NetworkAnalysis (MiNA [33]) toolset in ImageJ. Mitochondrial footprint was normalized to cell counts on the basis of DAPI staining of the corresponding images. Mitochondrial footprint, mean branch length, summed branch length, and network branches were compared using at least three images from two parallel stainings.

### 4.8. Calcium Signal Measurements

Before the calcium measurements, NPCs were seeded for two days onto eight-well chambers previously coated with poly-ornithine/laminin. Before measurement, NPCs cells were incubated in 1.0 μM Fluo4 AM (Thermo Fisher Scientific, Waltham, MA, USA) in a serum free culture medium for 15 min at 37 °C. Extracellular Fluo-4 AM was removed by changing the medium to Hanks’ balanced salt solution (Thermo Fisher Scientific, Waltham, MA, USA), supplemented with 20 mM Hepes (pH = 7.4) (Thermo Fisher Scientific, Waltham, MA, USA) and 0.9 mM MgCl2 (Sigma, St. Louis, MO, USA) (HBSS). All experiments were performed in HBSS at 37 °C maintained by Ibidi Heating System. The ligands were diluted to final concentrations in HBSS as well (glutamate (100 μM) (Sigma, St. Louis, MO, USA) and KCl (50 mM) (Sigma, St. Louis, MO, USA)). Time lapse sequences of cellular fluorescence images were recorded with the FluoViewTiempo (v. 4.3, Olympus, http://www.olympusmicro.com) software described earlier [34]. Fluorescence images were acquired between 505 and 525 nm at 488 nm excitation. Image analysis was carried out with ImageJ software from 4–5 independent experiments. The fluorescence data were normalized to the baseline signals (F/F0) where F is the integrated fluorescence intensity detected on cell bodies at a given time point and F0 is the average of integrated fluorescence intensities measured on the same cell bodies before admission of stimulants.

### 4.9. Statistics

All information related to statistical tests is documented in the corresponding figure legends and/or in the main text and in the Appendix A. Outlier data points were detected by Grubbs’s test and omitted from evaluations. Statistical comparisons for biological assays were performed in GraphPad Prism v. 8.00 for Windows (GraphPad Software, La Jolla, CA, USA, www.graphpad.com). We used two-way ANOVA statistics to evaluate the differences between the two differentiation methods and the donors. In case we found significant differences, we used Tukey’s multiple comparison test to detail the source of difference. An exception to this was the evaluation of RT-PCR results, where we compared mRNA expression levels using one-way ANOVA and Tukey’s multiple comparison test. Threshold for significance was set at 0.05.

## Figures and Tables

**Figure 1 ijms-21-09118-f001:**
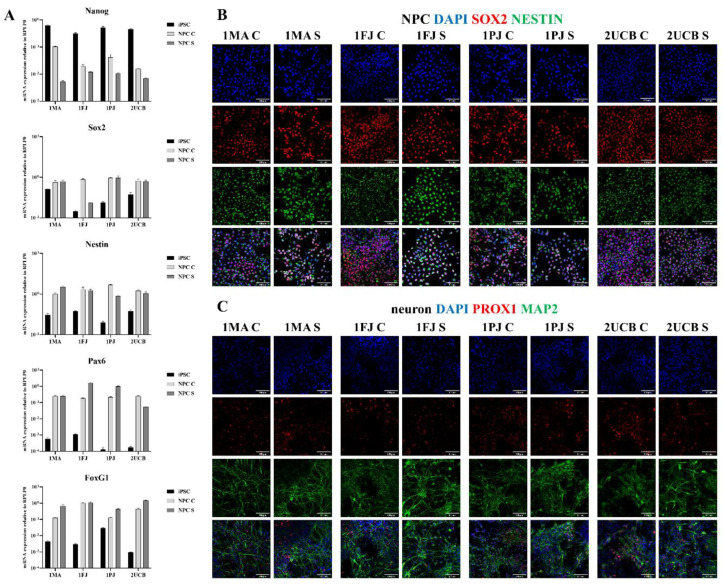
Protein and mRNA expression of state-specific markers: (**A**) relative mRNA expression levels to housekeeping gene, ribosomal protein lateral stalk subunit P0 (RPLP0), acquired in at least two biological parallel RT-PCR experiments (three technical replicates each). Pluripotency markers (NANOG and SOX2 (SRY (sex determining region Y)-box 2)), NPC marker (nestin and SOX2), and neural differentiation markers (paired box protein 6 (PAX6) and forkhead box protein G1 (FOXG1)) were investigated in induced pluripotent stem cells (iPSCs) and shortcut neural progenitor cells (NPCs). Values represent the means ± SEM. For statistical analysis see Appendix A. (**B**) Representative images of at least two parallel immunofluorescence stainings on the NPC cultures for NPC markers: Sox2 (red) and nestin (green); nuclei counterstained with DAPI (2-(4-amidinophenyl)-1*H*-indole-6-carboxamidine, blue) (scale bar 100 µm). (**C**) Representative images of at least two parallel immunofluorescence stainings on the neuron cultures for neural differentiation marker microtubule-associated protein 2 (Map2; green) and hippocampal granule cell-specific marker Prox1 (red); nuclei counterstained with DAPI (blue) (scale bar 100 µm).

**Figure 2 ijms-21-09118-f002:**
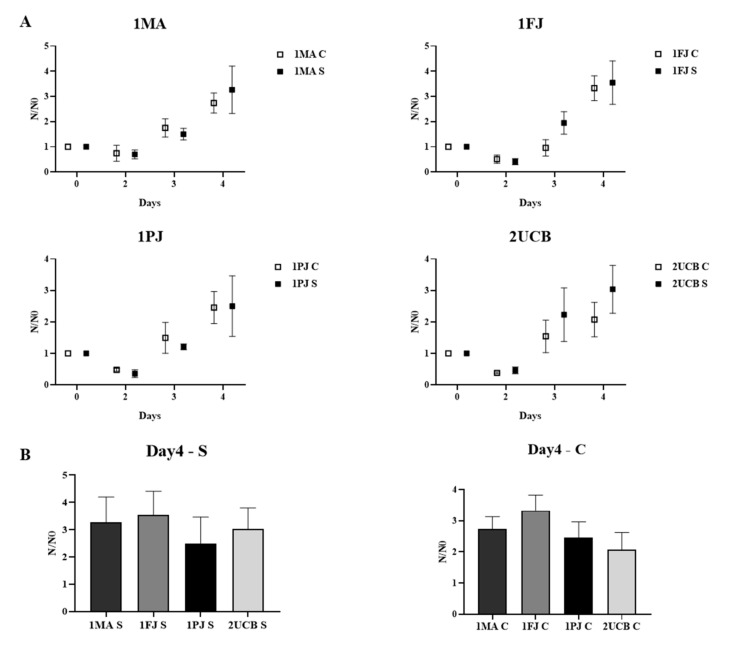
Comparison of the proliferative capacity of neural progenitor cells (NPCs) by flow cytometry (FCM) measurements. (**A**) Cell counts relative to seeded cell numbers are plotted per day; cNPCs and sNPCs derived from the same blood sample are placed side by side. (**B**) Cell counts (N) relative to seeded cell number (N0) on day 4 are compared between all cNPC lines and all sNPC lines. Values represent the means ±SEM of from at least 3 biological and 3–3 technical parallels.

**Figure 3 ijms-21-09118-f003:**
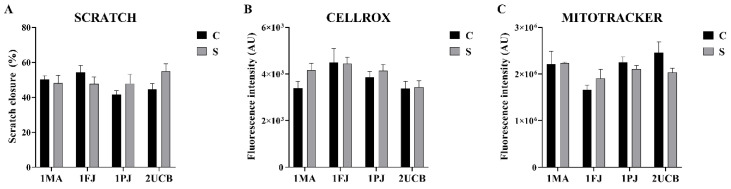
Functional analysis of neural progenitor cells (NPCs): (**A**) scratch assay. Average percentage of scratch closure calculated from two biological parallel experiments (three technical replicates each). Values represent means ± SEM. (**B**) Levels of reactive oxygen species (ROS) accumulation estimated by the CELLROX™ Green Reagent. Green fluorescence intensities—proportional to ROS levels—of NPCs after incubation with the reagent were measured by FCM (Flow Cytometry) with the exclusion of dead cells by propidium iodide staining. The averages of the medians of green fluorescence intensities ± SEM in at least three replicates are shown. (**C**) Mitochondrial functions compared by MitoTracker™ red reagent. NPCs—priory plated on 8-well confocal chambers—incubated with the reagent, were fixed by paraformaldehyde (PFA) and their nuclei were stained by DAPI (2-(4-amidinophenyl)-1*H*-indole-6-carboxamidine). Fluorescent images were taken in high content screening system on the whole surface of the chambers. Two biological parallels were evaluated (at least 25 frames each), and the averages of integrated red fluorescence intensities relative to cell counts and ±SEM are represented.

**Figure 4 ijms-21-09118-f004:**
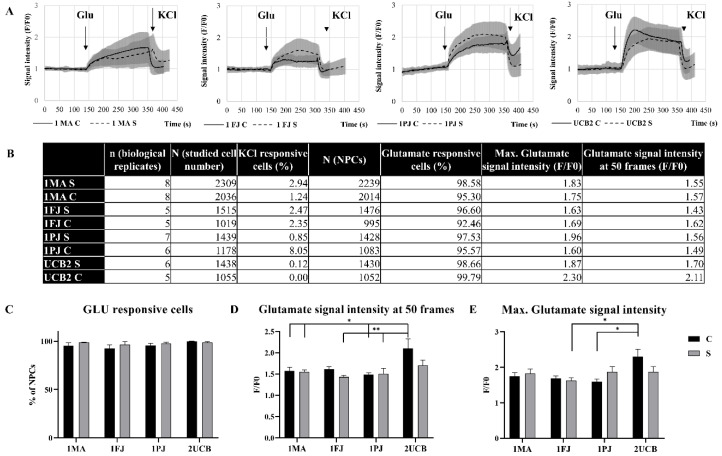
Functional characterization of neural progenitor cells (NPCs) by glutamate-induced calcium signals. Calcium levels were recorded by confocal microscopy in cells loaded with Fluo4-AM calcium indicator dye. F/F0 values were calculated, where F is the integrated fluorescence intensity detected on cell bodies at a given time point, and F0 is the average of integrated fluorescence intensities measured on the same cell bodies before the admission of stimulants. (**A**) Representative curves of calcium signals in NPC cultures triggered by glutamate (100 μM) and KCl (50 mM). One measurement (average and SD of at least 120 individual cell signals) of at least five biological replicates for each cell line are shown; cNPCs and sNPCs derived from the same blood sample are stacked side by side. (**B**) A comprehensive table of the calcium imaging data. (**C**) The average percentage ± SEM of glutamate responsive cells over all biological replicates. (**D**) Average of maximal fluorescence intensity ± SEM in 50 frames after glutamate admission over all biological replicates. (**E**) Average of maximal fluorescence intensity ± SEM after glutamate admission over all biological replicates. * *p* values <0.05; ** *p* values <0.01.

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
