# Peer review of "Functional Comparison of Blood-Derived Human Neural Progenitor Cells"

_ijms, 2020, doi:10.3390/ijms21239118_

Round 1

Reviewer 1 Report

In this manuscript, the authors reported a method to generate NPCs from blood mononuclear cells. The authors also compared the differences between two NPCs generated according to different protocols, including cNPC and sNPC. In addition, they obtained blood cells from a schizophrenia patient and two healthy family controls, and performed functional tests. Although the innovation in this study is sufficient, the inference of pathophysiology of schizophrenia cannot be definitively concluded, especially since the data is based on the data of a single patient, which will make the evidence relatively weak. In addition, the conclusions of Wnt and glutamate signaling are also very vague. I think it is necessary to provide more and convinced cell physiology data. Some specific comments are also attached below.

  1. Fig. 1: The images seem to be compared with the results of qPCR. However, compared with the results of qPCR, images lack the results of iPSC as a comparison. Therefore, it is difficult for us to understand how the actual protein expression changes after the two differentiation processes?
  2. Fig. 2a: Why does the number of cells seem to decrease slightly on the second day after seeding?
  3. Fig. 3B and 3C: The unit of the vertical axis is difficult to understand. What are "F medium" and "F/cell count"? In addition, in the part of Fig. 3C, the results were obtained through high content analysis, but the image analyzed by ImageXpress (Molecular device LLC) seems to be obtained by Zeiss confocal microscopy. Since these two systems are not coupled, in order to ensure the correctness of ImageXpress analysis and calculation, the authors should provide the mask images of his software for mitochondria judgment in the supplemental data to prove that the HCA parameter settings are correct.
  4. The data in Fig. 1 to Fig. 3 seem to have no statistical symbols? Is it because there is no difference, or is it not calculated? And, this is not clearly stated in legends.
  5. Although the authors seem to prove that cNPC and sNPC are actually very similar, there is a significant difference in the genes and time they need to be generated. However, what is the practical difference between the two? The authors should provide more information and in-depth discussion, especially to analyze the possible advantages and disadvantages of the two methods in clinical applications in the future.
  6. The author seems to have confirmed that the Schizophrenia patients selected in this study have certain genetic mutations. Do the healthy controls recruited in this study carry the same mutations corresponding to such specific genes? Authors should clearly state this.

Minor: numerous typo in the manuscript. For example, the word intensity was misspelled as “inensity” in Supplementary Tab 4. The content needs to be carefully corrected.

Author Response

We would like to thank Reviewer #1 for her/his constructive attitude and questions.

Below are the answers to her/his specific questions.

Reviewer 1

In this manuscript, the authors reported a method to generate NPCs from blood mononuclear cells. The authors also compared the differences between two NPCs generated according to different protocols, including cNPC and sNPC. In addition, they obtained blood cells from a schizophrenia patient and two healthy family controls, and performed functional tests.

Although the innovation in this study is sufficient, the inference of pathophysiology of schizophrenia cannot be definitively concluded, especially since the data is based on the data of a single patient, which will make the evidence relatively weak. In addition, the conclusions of Wnt and glutamate signaling are also very vague. I think it is necessary to provide more and convinced cell physiology data.

We understand that focusing on one family implicates several limitations. However, this was a unique opportunity to focus on specific genes and DNMs within one family. While it is difficult to generalize these findings to unrelated SZ patients and investigate cell-autonomous phenotypes associated with SZ in general, we believe that the approach of looking at alterations within genetically closely related members of a family meets the objectives of personalized or precision medicine. Furthermore, the main focus of this study was the pairwise comparison of differently generated NPC cultures derived from the same healthy or diseased donors, to demonstrate the applicability of both cell types in disease modelling.

Some specific comments are also attached below.

  1. Fig. 1: The images seem to be compared with the results of qPCR. However, compared with the results of qPCR, images lack the results of iPSC as a comparison. Therefore, it is difficult for us to understand how the actual protein expression changes after the two differentiation processes?

We did not intend to compare protein and mRNA expression data, as there is only a partial overlap in the gene products investigated. We compared mRNA expression of differentiation-specific genes in the differently generated NPC-s and the iPSC-s to show the significant difference between the NPCs’ and the iPSCs mRNA expressions, as well as to demonstrate the expression of neural progenitor genes (Nestin, Pax6, FoxG1) in both sNPCs and cNPCs. In the revised manuscript, to highlight this finding, we have rearranged Figure 1 and placed the PCR analysis as Panel A. Immunofluorescence stainings shown in Figure 1B are to demonstrate that neural progenitor markers, Nestin and SOX2 are expressed at similar levels in the differently generated NPCs (percentage of double positive cells were between 90-99% for all samples). Moreover, these images reveal similar morphologies of these cells. The detailed description of iPSCs, generated from blood samples of this family trio, has been sent as an independent technical report into Stem Cell Research and it is under review. We have indicated this as a reference. All data are available upon request.

average % double positive SD of % double positive

1MA S 96.03 1.01

1MA C 94.01 7.89

1FJ S 99.71 0.40

1FJ C 95.74 4.22

1PJ S 96.34 1.79

1PJ C 90.42 7.91

UCB2 S 99.73 0.38

UCB2 C 96.08 1.93

  1. Fig. 2a: Why does the number of cells seem to decrease slightly on the second day after seeding?

There is always a portion of the seeded cells that are unable to re-attach to the new growth surface after passage. In addition, we observed that there is a lag (about 24 hours) in cell proliferation after seeding. These facts could explain the cell deficit observed on day 2.

  1. Fig. 3B and 3C: The unit of the vertical axis is difficult to understand. What are "F medium" and "F/cell count"? In addition, in the part of Fig. 3C, the results were obtained through high content analysis, but the image analyzed by ImageXpress (Molecular device LLC) seems to be obtained by Zeiss confocal microscopy. Since these two systems are not coupled, in order to ensure the correctness of ImageXpress analysis and calculation, the authors should provide the mask images of his software for mitochondria judgment in the supplemental data to prove that the HCA parameter settings are correct.

In the revised manuscript, we have changed the unit of the vertical axis in Figures 3B and 3C for the sake of clarity. Now it is shown as “Fluorescence intensity (AU)”, and a detailed explanation is given in the figure legends.

We acquired fluorescence images of the same Mitotracker stainings by both ImageXpress High Content Screening system (Molecular Devices) and confocal microscope (Zeiss LSM 900). The images taken by the HCS system were evaluated by ImageXpress software, and the results were expressed as the integrated fluorescence divided by the cell number. The confocal images were evaluated by ImageJ software using NetworkAnalysis (MiNA) toolset as we indicated in the Materials and Methods section. We clarified the text accordingly in the results section.

“Cells were stained using the recommended protocol, and the fluorescence intensity of the dye was evaluated and normalized to cell numbers by ImageXpress High Content Screening system.

… In order to evaluate mitochondrial morphology, we analysed confocal images (Suplementary figure 3c) taken by Zeiss LSM 900, using the NetworkAnalysis (MiNA) toolset  of ImageJ software to determine the mitochondrial footprint, mean branch length, summed branch length, and network branches.”

  1. The data in Fig. 1 to Fig. 3 seem to have no statistical symbols? Is it because there is no difference, or is it not calculated? And, this is not clearly stated in legends.

We omitted statistical symbols on the graphs of Figure 1 for the sake of clarity. As we mentioned in the Results section, the supplementary Table 1 shows the statistical analysis of qPCR data and we have also referred to that in the figure description of the revised MS. In Fig. 3, there are no labels for significant differences, while we specified the significance (p values) in the body text of the manuscript.

  1. Although the authors seem to prove that cNPC and sNPC are actually very similar, there is a significant difference in the genes and time they need to be generated. However, what is the practical difference between the two? The authors should provide more information and in-depth discussion, especially to analyze the possible advantages and disadvantages of the two methods in clinical applications in the future.

We highlighted the advantages of the shortcut method in the discussion and the future perspectives sections. It is important to note that the method we describe here is for toxicology studies and disease modelling rather than for clinical applications.  In the revised manuscript, we made efforts to make this clear for the readers.

“Our results showed that classic and shortcuts NPCs have similar characteristics and the shortcut NPCs can be properly used for monitoring in vitro phenotypes. In this study, we demonstrated that bypassing the iPSC state reduces the cost and time thus makes the analysis feasible for a higher number of samples.”

  1. The author seems to have confirmed that the Schizophrenia patients selected in this study have certain genetic mutations. Do the healthy controls recruited in this study carry the same mutations corresponding to such specific genes? Authors should clearly state this.

The patient is a carrier of de novo mutations, and the parents do not have these mutations. The detailed description of iPSCs, generated from blood samples of this family trio, has been sent as an independent technical report into Stem Cell Research and it is under review, and referenced here. We have also included a sentence in the Method section: “The unrelated healthy control was not studied for these specific genetic mutations.”

Minor: numerous typo in the manuscript. For example, the word intensity was misspelled as “inensity” in Supplementary Tab 4. The content needs to be carefully corrected.

We have corrected the typos in the revised manuscript.

Reviewer 2 Report

The manuscript illustrates a comparison between two different methods to obtain Neural Progenitor Cells (NPCs) from Peripheral Blood Mononuclear Cells (PBMCs) of four subjects: one schizophrenic male patient and his unaffected parents and a healthy newborn female as control. Authors want to demonstrate that NPCs generated with the partially reprogramming of PBMCs avoiding the Induced Pluripotent Stem Cell (iPCS) stage show similar results compared to the fully reprogrammed ones in terms of growth, migration capacity, calcium signaling and oxidative stress. They assert that this shortcut reprogramming protocol provides an homogenous NPCs population and reduces time and costs concluding that it would be suited to study schizophrenia and for drug screening.

The paper is interesting but it raises some food for thoughts. I listed my concerns below recommending a major revision.

  • One of the issues of direct lineage reprogramming is the restricted scalability. Many studies highlighted that the absence of the pluripotent stage lead to an absence of a proliferative step, so the number of post-mitotic cells is finite. Authors should demonstrate that this method in principle allowing for the generation of sufficient cells quantities for downstream application. I mean, how long the induced NPCs could be expanded.
  • Yield and purity of the final population are two important parameters that could be calculated to evaluate the overall conversion efficiency of any cell reprogramming protocol. Please provide them.
  • Authors should justify why they use a Sendai virus vector (SeV) since Episomal Vectors (EV) are more advantageous than SeV; the improved EV is more efficient than SeV in reprogramming blood cells and the production of EV plasmids is more simple and affordable rather than generation of SeV vectors more challenging and expensive.
  • Authors should demonstrate the ability of NPCs to generate not only neurons but also glia cells as oligodendrocytes and astrocytes.
  • Authors did not specify if the PBMCs were fresh or previously frozen. If they were fresh, it would be interesting to know if this protocol would be applicable to short-term and long-term stored PBMCs at cold temperature.
  • Finally, to test the general applicability of the protocol the number of PBMCs donors should be increased and diversified.

Author Response

 We would like to thank Reviewer #2 for the thorough review of our work and valuable remarks.

Reviewer 2

The manuscript illustrates a comparison between two different methods to obtain Neural Progenitor Cells (NPCs) from Peripheral Blood Mononuclear Cells (PBMCs) of four subjects: one schizophrenic male patient and his unaffected parents and a healthy newborn female as control. Authors want to demonstrate that NPCs generated with the partially reprogramming of PBMCs avoiding the Induced Pluripotent Stem Cell (iPCS) stage show similar results compared to the fully reprogrammed ones in terms of growth, migration capacity, calcium signaling and oxidative stress. They assert that this shortcut reprogramming protocol provides an homogenous NPCs population and reduces time and costs concluding that it would be suited to study schizophrenia and for drug screening.

The paper is interesting but it raises some food for thoughts. I listed my concerns below recommending a major revision.

  • One of the issues of direct lineage reprogramming is the restricted scalability. Many studies highlighted that the absence of the pluripotent stage lead to an absence of a proliferative step, so the number of post-mitotic cells is finite. Authors should demonstrate that this method in principle allowing for the generation of sufficient cells quantities for downstream application. I mean, howlong the induced NPCs could be expanded.

As we demonstrated in Figure 2. NPCs grow relatively fast, thus the proliferative state is present in these cases, while both classical and shortcut NPCs show signs of senescence, that is slower proliferation rate and altered morphology after 35 passages. We included a sentence related to this into the Methods and Discussion sections of the revised manuscript.

“Both sNPCs and cNPCs could be propagated up to 35 passages, when they started to show signs of senescence (i.e. slower proliferation rate and altered morphology). cNPCs between passage p6 and p18 and sNPCs between passage p10 and p18 were used for the experiments.”

“Both types of NPCs can be propagated at least passage 30 without losing their proliferative capacity providing sufficient cells quantities for functional assays and drug screening applications.”

  • Yield and purity of the final population are two important parameters that could be calculated to evaluate the overall conversion efficiencyof any cell reprogramming protocol. Please provide them.

As the NPCs proliferate fast (at least until passage 30), we obtained several times greater cell numbers for each passage than the initial cell number, thus, we did feel no need for calculating the yield. As to the purity and cell conversion efficiency, we have performed a quantitative analysis of Nestin-SOX2 stainings, and found that both classical and shortcut NPCs contained similar amounts of double positive cells (between 90-99%) for each donor.

                 average % double positive   SD of % double positive

1MA  S      96.03                                     1.01

1MA C       94.01                                     7.89

1FJ S         99.71                                     0.40

1FJ C        95.74                                     4.22

1PJ S         96.34                                    1.79

1PJ C        90.42                                      7.91

UCB2 S      99.73                                    0.38

UCB2 C      96.08                                    1.93

  • Authors should justify why they use a Sendai virus vector (SeV) since Episomal Vectors (EV) are more advantageous than SeV; the improved EV is more efficient than SeV in reprogramming blood cells and the production of EV plasmids is more simple and affordable rather than generation of SeV vectors more challenging and expensive.

According to the literature; “The Cytotune SeV reprogramming kit has achieved a high acceptance rate for skin fibroblast and blood cell reprogramming; we share this enthusiasm and highly recommend SeV reprogramming to laboratories that do not focus on generating clinical-grade hiPSCs.” (1), thus we have chosen SeV technique when we started this work in 2016. EV vectors are more advantageous for cell therapy purposes; however we are not convinced that this is also the case for disease modelling (2). Comparison of the two methods is beyond the scope of this study. We have justified in the introduction why we used the Sendai virus vector as suggested by the reviewer.

“To allow such a comparison, we used Sendai virus reprogramming method for the generation of NPCs from fully reprogrammed (classic) and partially reprogrammed (shortcut) cells, because this method allows to generate the desired cell types effectively without transgene integration [10,11].”

  • Authors should demonstrate the ability of NPCs to generate not only neurons but also glia cells as oligodendrocytes and astrocytes.

We compared our shortcut NPCs to a type of hippocampalNPCs which are the precursors of the Dentate Gyrus Prox1 positive granule cells, thus we focused on neuronal differentiation. We agree with the Reviewer that generation of glial cells from both types of NPCs and a functional analysis of them would be interesting, while this is beyond the scope of the present study.

  • Authors did not specify if the PBMCs were fresh or previously frozen. If they were fresh, it would be interesting to know if this protocol would be applicable to short-term and long-term storedPBMCs at cold temperature.
  • Finally, to test the general applicability of the protocol the number of PBMCs donors should be increased and diversified.

We have used freshly isolated PBMC samples in the recent study and according to the Reviewer’s suggestion we have indicated this in the Methods section of the revised manuscript.

“iPSC lines were generated from freshly isolated PBMCs (using BD Vacutainer CPT Cell Preparation Tubes with Sodium HeparinN. REF 362753) of healthy and diseased individuals by Sendai virus”

In a separate study we have successfully established iPSCs and shortcut NPCs from a previously frozen PBMC sample parallel with fresh blood samples (3 samples). We found no difference between the two types of samples (fresh vs frozen) in terms of success of generation of NPCs or the NPC marker expression. Since it is an on-going project thus far we only have limited data about differentiation capacity and functional assays. Nevertheless, our preliminary data showed that this method can also be used for generation NPCs from frozen blood samples.

We agree with the reviewer that generalization of the applicability needs involvement of a high number of PBMC donors with various backgrounds, as well as engagement of numerous research centres. We believe that our recent work helps to provoke such multicentre study in the near future.

References:

1. A comparison of non-integrating reprogramming methods.Schlaeger TM, Daheron L, Brickler TR, Entwisle S, Chan K, Cianci A, DeVine A, Ettenger A, Fitzgerald K, Godfrey M, Gupta D, McPherson J, Malwadkar P, Gupta M, Bell B, Doi A, Jung N, Li X, Lynes MS, Brookes E, Cherry AB, Demirbas D, Tsankov AM, Zon LI, Rubin LL, Feinberg AP, Meissner A, Cowan CA, Daley GQ.Nat Biotechnol. 2015 Jan;33(1):58-63. doi: 10.1038/nbt.3070. Epub 2014 Dec 1.

2. An insight into non-integrative gene delivery approaches to generate transgene-free induced pluripotent stem cells.Haridhasapavalan KK, Borgohain MP, Dey C, Saha B, Narayan G, Kumar S, Thummer RP.Gene. 2019 Feb 20;686:146-159. doi: 10.1016/j.gene.2018.11.069. Epub 2018 Nov 22.PMID: 30472380 Review.

Round 2

Reviewer 1 Report

I agree with all the replies made by the authors. Therefore, recommending this manuscript can be considered accepted for publication.

Reviewer 2 Report

Authors answered to all my concerns. Manuscript can be published in the present form.

This manuscript is a resubmission of an earlier submission. The following is a list of the peer review reports and author responses from that submission.

Round 1

Reviewer 1 Report

This is a very good study. The authors report a method to successfully shortcut neural progenitor cells from blood mononuclear cells. Blood cells were obtained from a schizophrenia patient, the healthy parents of the patient, and a healthy newborn. Several functional tests were carried out of pathways, which are related to schizophrenia pathology (glutamater signaling, Wnt signaling etc).

Minor points:

  1. Please improve the quality of Suppl. Fig. 2A (microphotographs).
  2. line 83: can be overcomed (not overcome).
  3. line 86: immature relatives (not relative).

Author Response

We appreciate the positive evaluation of our manuscript and thank the reviewer. In the following paragraphs we answer her/his specific questions.  

Minor points:

  1. Please improve the quality of Suppl. Fig. 2A (microphotographs).
  • In the revised manuscript (RM) we have improved the quality of the microphotographs.
  1. line 83: can be overcomed (not overcome).
  • In the revised manuscript (RM) we have corrected the sentence.
  1. line 86: immature relatives (not relative).
  • In the revised manuscript (RM) we have corrected the sentence.

Reviewer 2 Report

NPCs derived from somatic cells is a powerful tool in studying genetic diseases, for instance, familial pathologies caused by germline mutations. It hasn’t been illustrated the genetic causation of the Schizophrenia disease case utilised in the current work. Neither has been mentioned the genetic profile of the parents from the family trio. Since Schizophrenia can be complexed by multiple inherited or acquired attributors, it is not clear from the beginning wether the “patient” line can be applied as a real representation of the disease model. Both disease and control cases should have been genotyped in the first instance. 

The expression level of cell type specific markers should have been accurately quantified (e.g, western-blotting, RT-PCR), in addition to fluorescence microscopy.

It should have been explained the time span (e.g. number of days) needed for each single step of NPC and neuronal differentiation. A schematic diagram would help explain the differentiation process.  

Line198 "mitochondria in the NPCs. The cells were stained, and the fluorescence intensity of the dye was evaluated”:Mitotracker has been widely used to visualise the distribution and dynamic of mitochondrial network. The intensity of Microtracker fluorescence doesn’t necessarily indicate the cellular fitness. Instead, a workflow should have been established to analyse mitochondrial morphology (e.g. footprint, volume, fragmentation) using fluorescence images.

Line 182 "patient-derived NPCs showed higher ROS levels (Figure 3b) than the controls in both NPC groups, while there was no significant difference between the differently generated NPCs (p=0.2233 by one 184 way ANOVA statistic)" and  Line 211 "from further analysis. We have analysed hundreds of cells (Figure 4b), and found no significant differences in the number of responding cells or signal intensities after addition of glutamate, when comparing the patient specific NPCs to NPCs derived from his parents (Figure 4c,d and e)”: For most assays, cNPCs and sNPCs don’t show significant difference in mean, it has therefore been concluded the similar performance between iPSC-derived method and the shortcut protocol. However, it is noticed a generally high SD between experimental groups which might be a main confounder. Given the modest variability between disease and control groups, it is believed that further validation is need for any confident conclusion. 

Figure 1: the resolution of all images is too low to distinguish even between cells. There is no information in statistical significance.

Figure4: e, f panels overlapping

Author Response

We would like to thank Reviewer 2 for the supportive attitude and valuable questions. We have summarized our answers and rebuttals below.

Reviewer 2

NPCs derived from somatic cells is a powerful tool in studying genetic diseases, for instance, familial pathologies caused by germline mutations. It hasn’t been illustrated the genetic causation of the Schizophrenia disease case utilised in the current work. Neither has been mentioned the genetic profile of the parents from the family trio. Since Schizophrenia can be complexed by multiple inherited or acquired attributors, it is not clear from the beginning wether the “patient” line can be applied as a real representation of the disease model. Both disease and control cases should have been genotyped in the first instance.

  • The SCZ patient was selected from a whole exome sequencing study, where case-parent trios were analysed for de novo mutations. The detailed data of the patient selection and the iPSC generation from this trio is under review in Stem Cell Research. However we have now included a paragraph into Materials and Methods section (lines 310-314) according to the suggestion of Reviewer 2:

“The iPSCs were established from a male schizophrenia patient, who is a carrier of 3 nonsynonymous de novo mutations  in genes leucine rich repeat containing 7 (LRRC7 (1:70505093G>A)), KH-Type Splicing Regulatory Protein (KHSRP (19:6416869C>A)), and Killer Cell Immunoglobulin-Like Receptor, Two Domains, Long Cytoplasmic Tail, 1 (KIR2DL1(19:55286658A>T)) and from his unaffected parents (Hathy et al. under review).”           

The expression level of cell type specific markers should have been accurately quantified (e.g, western-blotting, RT-PCR), in addition to fluorescence microscopy.

  • In this paper we focused on the NPCs generated by two different methods. In Fig 1 we wanted to highlight that these NPCs express early neural markers Nestin and SOX2, and can be differentiated into PROX1 and MAP2 double positive neurons. According to the suggestion of Reviewer 2, we now included an RT-PCR analysis of SOX2 as an early neural marker, along with Nestin, PAX6 and FOXG1. However, a detailed analysis of the differentiation into PROX1-expressing neurons is beyond the aims of the present manuscript, and this would require optimization of the differentiation protocol for each cell line to obtain the highest efficacy, as well as a functional characterization of these selected neurons.

We have improved the quality of ICC images and rearranged the figure 1c including SOX2, and an additional sentence was inserted into the MS (lines 141-142), as well as in the corresponding Figure legends:

“SOX2 is known to be expressed in both iPSCs and NPCs, and its expression level varied within one order of magnitude in all these cell types.”

We also included the TaqMan® assay used in the gene expression section of Materials and Methods (line 388 ):

“SOX2 as undifferentiated and NPC marker;”

It should have been explained the time span (e.g. number of days) needed for each single step of NPC and neuronal differentiation. A schematic diagram would help explain the differentiation process.

  • As a detailed directed differentiation protocol of iPSCs into DG neurons has already been published [1] we have just summarized the time line and differentiation steps in Supplementary Figure 1. According to the recommendation of the Reviewer, we have rearranged Supplementary Figure 1. We have highlighted the time span and represent the actual time at the arrows. If it improves our manuscript, we can use this as graphical abstract as well.

Line198 "mitochondria in the NPCs. The cells were stained, and the fluorescence intensity of the dye was evaluated”:Mitotracker has been widely used to visualise the distribution and dynamic of mitochondrial network. The intensity of Microtracker fluorescence doesn’t necessarily indicate the cellular fitness. Instead, a workflow should have been established to analyse mitochondrial morphology (e.g. footprint, volume, fragmentation) using fluorescence images.

  • We have used MitoTracker Red CMXRos that stains mitochondria in live cells, its accumulation and the visualisation of the mitochondria depend on membrane potential. We have used this mitochondrial dye, because assays monitoring mitochondrial membrane potential are commonly used for evaluation of cellular fitness [2]. As the dye is well-retained after aldehyde fixation we also took confocal microscopy pictures, and in agreement with the Reviewer’s suggestion, performed an analysis of mitochondrial morphology using ImageJ software and an already published macro [3]. Our new data correlate well with our previous findings in NPCs, and are now presented in the revised Supplementary Figure 3d , with appropriate explanation in the MS (in the Results section (lines 213-223):

“Next, Mitotracker Red CMXRos (a mitochondrion-specific dye) was used for the quantification of mitochondria in the NPCs. Cells were stained using the recommended protocol, and the fluorescence intensity of the dye was evaluated and normalized to cell numbers. There were no significant differences between the NPCs (p=0.3966 by one way ANOVA statistic), showing the similarity of NPCs derived by the two different methods. In order to evaluate mitochondrial morphology, we have analysed confocal images to determine the mitochondrial footprint, mean branch length, summed branch length and network branches. We found no significant differences between mitochondrial footprints (p=0.2176), and mean branch length (p=0.1004). For summed  branch length and network branches we found no significant difference between NPCs derived from the patient and the control NPCs, while significant differences were found between 1FJ NPC S and UCB2 NPC C (p= 0.0334and p= 0.0122, respectively). ”

We included this method in the Mitotracker assay section of Materials and Methods (lines 424-428):

“For mitochondrial morphology analysis 3 parallel confocal images were evaluated for each cell lines using the Mitochondrial NetworkAnalysis (MiNA- [30]) toolset in ImageJ. Mitochondrial footprint was normalized to cell counts based on DAPI staining of the corresponding images. Mitochondrial footprint, mean branch length, summed branch length and network branches were compared using ANOVA statistics.”

Line 182 "patient-derived NPCs showed higher ROS levels (Figure 3b) than the controls in both NPC groups, while there was no significant difference between the differently generated NPCs (p=0.2233 by one 184 way ANOVA statistic)" and  Line 211 "from further analysis. We have analysed hundreds of cells (Figure 4b), and found no significant differences in the number of responding cells or signal intensities after addition of glutamate, when comparing the patient specific NPCs to NPCs derived from his parents (Figure 4c,d and e)”: For most assays, cNPCs and sNPCs don’t show significant difference in mean, it has therefore been concluded the similar performance between iPSC-derived method and the shortcut protocol. However, it is noticed a generally high SD between experimental groups which might be a main confounder. Given the modest variability between disease and control groups, it is believed that further validation is need for any confident conclusion.

  • We appreciate this comment - indeed, there are high SD values in some cases which might be a result of the complexity of live cells, or because many biological processes are intrinsically stochastic. We standardized our experiments using cells with the same starting cell number and similar passage numbers. However, working with a transient cell population (NPCs), which is heterogeneous and becomes senescent or partly differentiated in time, gives rise to significant variability. In general, increasing the number of measurements also can decrease the resulting SD and p value. Therefore we included more data points (when it was applicable in this short period of time) and recalculated the SD and p values. This did not considerably change the results. Still, significant differences in the proliferation rate, Ca2+ signals evoked by glutamate (at 50 frames) or mitochondrial morphology were seen, despite the high SD values. Therefore we have concluded that shortcut NPCs can be useful for monitoring in vitro phenotypes.

Figure 1: the resolution of all images is too low to distinguish even between cells. There is no information in statistical significance.

  • In accordance with this suggestion, we have improved the resolution of Figure 1 and calculated the statistical significance. In Figure 1 we wanted to demonstrate that the differentiation into neural progenitors and neurons can be achieved by both methods. We performed RT-PCR analysis at two different passage numbers (to make sure that we are working with proper neural progenitors), thus the strength of statistical analysis is relatively low. However we included a two way Anova analysis in the manuscript (see Supplementary table 1) as Reviewer 2 suggested, and inserted a sentence about this in the MS (lines 144-147 ).

“The statistical analysis showed significant differences between iPSCs and NPCs (TableS1), but the differences were not significant between two types of NPCs (except the FJ where NPCs resulted by the classical differentiation showed lower Pax6 and FoxG1 expression than the sNPCs).”

Figure4: e, f panels overlapping

  • In the revised manuscript we replaced Figure 4 with the corrected one.

  1. Yu DX, Di Giorgio FP, Yao J, Marchetto MC, Brennand K, Wright R, Mei A, McHenry L, Lisuk D, Grasmick JM, et al. (2014) Modeling hippocampal neurogenesis using human pluripotent stem cells. Stem cell reports 2: 295-310
  2. Rasola A, Geuna M (2001) A flow cytometry assay simultaneously detects independent apoptotic parameters. Cytometry 45: 151-157
  3. Valente AJ, Maddalena LA, Robb EL, Moradi F, Stuart JA (2017) A simple ImageJ macro tool for analyzing mitochondrial network morphology in mammalian cell culture. Acta histochemica 119: 315-326

Round 2

Reviewer 2 Report

The work has been well strengthened - Agree for publication.